



# The Dependence of Aerosols' Global and Local Precipitation Impacts on Emitting Region

Geeta G. Persad[1]

[1]Department of Geological Sciences, The University of Texas at Austin, Austin, Texas 78712 USA

*Correspondence to*: Geeta G. Persad (geeta.persad@jsg.utexas.edu)

**Abstract.** The influence of aerosol emissions' geographic distribution on the magnitude and spatial pattern of their precipitation impacts remains poorly understood. In this study, the NCAR CESM1 global climate model is used in coupled atmosphere-slab ocean mode to simulate the global hydrological cycle response to a fixed amount and composition of aerosol emitted from 8 key source regions. The results indicate that the location of aerosol emissions is a strong determinant of both

the magnitude and spatial distribution of the hydrological response. The global-mean precipitation response to aerosol emissions is found to vary over a six-fold range depending solely on source location. Mid-latitude sources generate larger global-mean precipitation responses than do tropical and sub-tropical sources, driven largely by the formers' stronger global-mean temperature influence. However, the spatial distribution of precipitation responses to some (largely tropical and sub-tropical) regional emissions are almost entirely localized within the source region, while responses to other (primarily mid-

latitude) regional emissions are almost entirely remote. It is proposed that this diversity arises from the differing strength with which each regional emission generates fast precipitation responses that remain largely localized. The findings highlight that tropical regions are particularly susceptible to hydrological cycle change from either local or remote aerosol emissions, encourage greater investigation of the processes controlling localization of the precipitation response to regional aerosols, and demonstrate that the geographic distribution of anthropogenic aerosol emissions must be considered when estimating their

hydrological impacts.

## 1 Introduction

The geographic distribution of anthropogenic aerosol emissions has evolved continuously over the industrial era and is projected to continue to do so. Emissions have transitioned from a locus in Western Europe and North America in the late 19th and early 20th century to South and East Asia in the present-day due to changing patterns of industrialization and air quality

regulation (Hoesly et al., 2018). Future projections of aerosol emissions, while highly uncertain (Samset et al., 2019), contemplate new growth of emissions in regions like South America and East Africa as industrialization of these regions accelerates (Lund et al., 2019).

With this redistribution of aerosol emissions comes the potential redistribution of aerosols' impact on the hydrological cycle, which are known to be substantial. Aerosols have been shown to have strong in-situ and remote hydrological cycle

impacts via their influence on the dynamics, thermodynamics, and microphysics that control precipitation (Boucher et al.,



2013). AAs' rapid spatiotemporal evolution over the 20th and early 21st century and its effect on the large-scale circulation has been identified as the dominant driver of an observed southward shift in the tropical Pacific rain belt (Allen et al., 2015) and the collapse and recent recovery in Sahel precipitation (Marvel et al., 2020). Observed weakening of the South and East Asian monsoons, meanwhile, has been attributed to both large-scale and local-scale aerosol forcing (Bollasina et al., 2011,
2014; B. Dong et al., 2019; Li et al., 2016; Singh et al., 2019).

A range of studies have established that the precipitation response to aerosols is dependent on the spatial distribution of aerosol forcing. The ongoing evolution in the spatial distribution of global aerosol emissions has been associated with an evolution in the spatial pattern of the corresponding global precipitation response (Deser et al., 2020; Kang et al., 2021). A range of studies isolating the response to historical emissions in individual regions or latitude bands have identified common
underlying features of this geographic dependence. Multi-model and single-model studies applying both idealized and historical regional aerosol perturbations generally find that higher latitude aerosol sources (Europe, North America) produce a stronger global-mean precipitation response than lower latitude sources (East or South Asia) when normalized by radiative forcing or atmospheric concentration (Kasoar et al., 2018; Liu et al., 2018; Shindell et al., 2012; Westervelt et al., 2018). However, some studies show strongly differing spatial distributions of precipitation response (Ishizaki et al., 2013; Liu et al.,
2018; Shindell et al., 2012), while others argue that the spatial patterns of response are similar regardless of source region (Kasoar et al., 2018).

Identifying the specific role of emission location in the climate response to aerosol, however, remains difficult based on the existing literature. Previous studies use actual or scaled historical emissions, which are unequal across regions, latitude bands, and/or time periods. Responses can be normalized by emissions, concentrations, or forcing to approximate the role of
source location. For example, Shindell et al. (2012) proposed and estimated a set of "Regional Precipitation Potentials" based on their analysis of the response to historical aerosol emissions in separate latitude bands, which quantifies the precipitation response per unit of radiative forcing for aerosol emissions in a given latitude band. However, these approaches assume linearity in the response to different amounts of aerosol. Additionally, existing studies tend to sample the effects of a relatively small subset of regions (generally confined to South and East Asia, North America, and Europe). Given the high uncertainty
and potential for growth in aerosol emissions within individual regions outside of this subset (Lund et al., 2019), understanding the relative importance of aerosol emissions from a larger range of regions may be potentially beneficial for near-term climate prediction as well as for fundamental understanding of the climate system response to heterogeneous forcing.

In Persad and Caldeira (2018), we designed a set of simulations in a coupled atmosphere-slab ocean global climate model (GCM) to evaluate the global-scale temperature response to identical aerosols (equal to year 2000 Chinese sulfate, black
carbon, and organic carbon emissions) emitted from 8 different regions and found a 14-fold range in the global-mean temperature response due solely to differences in emission source location. By fixing the amount of aerosol based on a historical reference, but varying the source location, the simulations isolate the role of the geographic location of emissions in setting the climate response to aerosols within a quasi-realistic framework. This strategy is analogous to Green's function approaches, in which a climate model is perturbed with an identical anomaly in several different locations one-by-one, that



have been used to evaluate radiative feedbacks (Dong et al., 2019; Zhou et al., 2017). In Persad & Caldeira (2018), the anomaly—i.e. a fixed emission of aerosol—is structured to match the national boundaries along which the policy and technological shifts that determine aerosol trends typically occur (O'Neill et al., 2015; Rao et al., 2017; Riahi et al., 2017). The goal is to achieve an experimental set-up that provides fundamental physical insight within a framework that is directly translatable to policy-relevant tools like emulators, emissions metrics, or social cost calculations that scale based on national

emissions. This allows an approach for understanding the importance of aerosols' geographic distribution to their climate response that is complementary to the unequal historical emissions or highly idealized forcing experiment designs that have previously been pursued.

In this study, the simulations from Persad and Caldeira (2018) are analyzed to understand the dependence of the hydrological cycle response to aerosols on emission source location. Section 2 describes the simulations and analysis

techniques used to assess the influence of identical aerosol emissions from different regions on global precipitation. Section 3 presents results assessing impacts on both global-mean precipitation and the spatial distribution of precipitation and explores a potential theory for why certain source regions produce strongly localized precipitation responses and others do not. Section 4 places the findings in the context of existing understanding, and Section 5 summarizes and explores the implications of the results in the context of the continuing spatial redistribution of anthropogenic aerosol emissions.

**2 Methods**

Simulations designed to test the global climate response to identical aerosol emissions in different source regions are conducted in the National Center for Atmospheric Research Community Earth System Model version 1.2 (CESM1) using the Community Atmosphere Model version 5 (CAM5) coupled to the Community Land Model version 4 (CLM4) and a mixed-layer ocean (Hurrell et al., 2013). The three lognormal mode (MAM3) modal aerosol module is used in CAM5, which allows

for interactive transport, growth, internal mixing, and removal of aerosol emissions by the internal physics of the model (Liu et al., 2012). This version of the model has been shown to produce minimal (<10%) biases in aerosol concentrations and aerosol radiative forcing compared to more complicated atmospheric chemistry models (Ghan et al., 2012; Liu et al., 2012). Simulations in CESM1(CAM5) using historical aerosol emissions compare well with observations of spatial and temporal patterns of aerosol optical depth (AOD), though low biases in AOD are apparent over South and East Asia. CAM5 allows for

simulation of aerosol-cloud interactions within the model's two-moment microphysical parameterization (Liu et al., 2012). Microphysical aerosol-precipitation interactions are permitted within the stratiform cloud microphysics representation, but are excluded from the cumulus cloud parameterization (Ghan et al., 2012). The fully coupled CESM1 consistently performs among the top 10 Fifth Coupled Model Intercomparison Project (CMIP5) generation models in simulation of historical temperature and precipitation trends and spatial patterns (Koutroulis et al., 2016).

The simulation suite consists of a control simulation and 8 perturbation experiments. In the control simulation, year 2000 conditions are imposed for all external and internal forcers, with the exception of non-biomass burning anthropogenic





emissions of black carbon, organic carbon, sulphur dioxide, and sulphate. These are set to 1850 values using CAM5's standard historical emissions fields (Lamarque et al., 2010). In the 8 perturbation experiments, these anthropogenic aerosol emissions fields are modified only within the relevant region to impose an additional total annual emission of black carbon, organic

carbon, and sulphate precursor equivalent to China's total year 2000 values (1.61 Tg, 4.03 Tg, and 22.4 Tg, respectively). This is achieved by scaling that region's year 2000 CAM5 standard historical emissions fields at each grid point by a fixed factor such that the total change in anthropogenic aerosol emissions between each of the 8 perturbation experiments and the control is identical. The spatial distribution of the emission perturbation thus follows the realistic year 2000 spatial pattern within a given region (Figure A1). The 8 perturbation regions chosen (Brazil, China, East Asia, India, Indonesia, South Africa, the

U.S., and Western Europe) are selected to sample a range of past, present, and projected future major emission source regions as well as a range of climate regimes (e.g. tropical, monsoonal, and extratropical in both hemispheres). A comparable set of simulations are run in atmosphere-only mode with sea surface temperatures (SSTs) and sea ice fixed. Further discussion of simulation characteristics and behaviour can be found in Persad & Caldeira (2018).

The atmosphere-only simulations are used to calculate effective radiative forcing (ERF) and atmospheric absorption and

to decompose the fast and slow precipitation responses. ERF is calculated as the change in global-mean top-of-atmosphere energy balance in each of the 8 atmosphere-only perturbation experiments compared to the atmosphere-only control, following the standard fixed SST definition of ERF (Ramaswamy et al., 2018). The fast precipitation response is calculated as the precipitation response within the atmosphere-only simulation. Atmospheric absorption values shown are also calculated within the atmosphere-only simulation. The slow precipitation response is calculated as the residual of the precipitation response in

the coupled simulations minus the fast precipitation response.

Model output is produced at a nominal 2-degree longitude by latitude resolution. Coupled simulations are run for 100 years, and atmosphere-only simulations are run for 60 years. The initial 40 (coupled simulations) or 20 (atmosphere-only simulations) years are treated as the transient response (based on analysis of trends in the top-of-atmosphere energy imbalance) and discarded. Statistical significance is estimated using either the last 60 years (coupled simulations) or 40 years (atmosphere-

only) of the simulations as the sample, with effective sample size adjusted to account for autocorrelation. Standard errors are provided for global-mean values. Statistical significance for maps is estimated using a two-tailed $t$ test, and the 95% confidence interval is reported.

**Figure 1.** Spatial distribution of the change in precipitation rate (mm/day) due to the addition of an identical amount and composition of aerosol emissions in each of the 8 regions. Gridlines indicates regions where the changes are not statistically significant at the 95% confidence level via a two-sided *t* test. Global mean precipitation rate change and standard error in μm/day are shown in the bottom left corner of each map.





## 3 Results

### 3.1 Global-mean precipitation response

The results indicate that differences in source location alone can produce a more than six-fold difference in global-mean precipitation response to aerosol emissions (Figure 1). Aerosols from all regions decrease global-mean precipitation. However, Western European emissions produce by far the strongest global-mean precipitation reduction (-21.3 ± 1.8 µm/day or an approximately 1% decrease)—a full 50% larger than the next strongest precipitation response (to U.S. emissions), while South Asian emissions produce the weakest (-2.6 ± 1.8 µm/day or <0.2% decrease). In general, mid-latitude sources (Western Europe, the U.S., China, and South Africa) generate larger global-mean precipitation responses than do the tropical and sub-tropical sources.

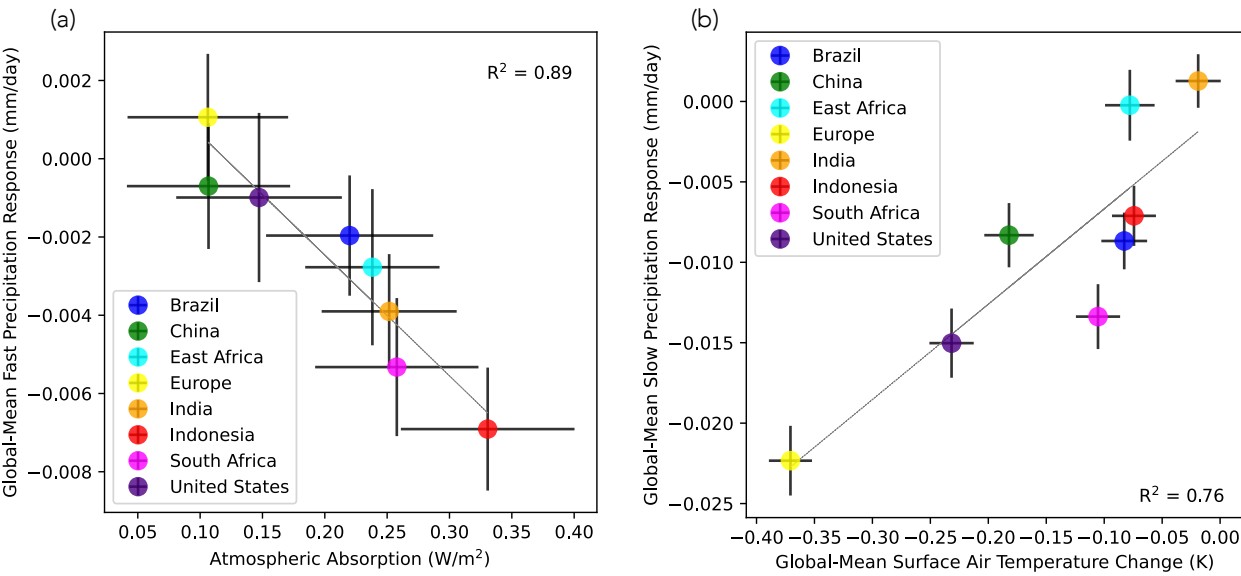

**Figure 2.** The total global-mean precipitation response to emissions from each region can be decomposed in the (a) global-mean fast precipitation response (mm/day, y-axis), which is strongly correlated with the global-mean change in atmospheric absorption (W/m², x-axis), and the (b) global-mean slow precipitation response (mm/day, y-axis), which is strongly correlated with the global-mean change in surface temperature (K, x-axis).

Global-mean precipitation changes can be separated into a fast (atmosphere-only) portion that scales strongly with the change in global-mean atmospheric absorption (Figure 2a) and a slow (atmosphere-ocean coupled) portion that scales strongly with the coupled change in global mean temperature (Figure 2b). Increased atmospheric absorption of radiative energy under fixed sea surface temperature conditions enhances atmospheric stability, suppressing evaporation, convection, and precipitation—producing the so-called "fast" precipitation response (Andrews et al., 2010; Dagan et al., 2019, 2021). Once sea surface temperatures are allowed to respond, feedbacks in moist convection and horizontal moist transport and thermodynamic





constraints on precipitation and evaporation result in a slow precipitation response that is estimated as a 2-3% increase in precipitation per degree Kelvin of global-mean warming, regardless of forcing (Samset et al., 2016; Sillmann et al., 2017). For most emitting regions simulated in this study, the slow precipitation response contributes the majority of the total precipitation response (Figure 3). The exceptions are East African and Indian emissions, whose total global-mean precipitation response is

almost entirely contributed (or, in the case of Indian emissions, outpaced) by the fast precipitation response, and Indonesian emissions, whose total precipitation response results from roughly equal slow and fast precipitation responses.

The diversity in the slow precipitation response, which in turn drives the majority of the diversity in the total precipitation response, is largely explained by the divergence in global-mean temperature responses (Figure 2b). This temperature

dependence also provides an explanation for the larger total global-mean precipitation responses generated by higher latitude sources. The temperature response to identical aerosols emitted from these source regions spans a 14-fold range. As detailed in Persad and Caldeira (2018), the differences in global-mean temperature response stem from a combination of the differing strengths of effective radiative forcing generated by the individual source regions and the different ability of the forcing to generate climate feedbacks (see Persad and Caldeira (2018), Figure 3b). The higher latitude source regions generally produce

larger effective radiative forcing than the lower latitude sources. Forcing from higher latitude sources, in turn, is also more effective at generating cloud and sea ice feedbacks that amplify its efficacy at generating temperature change relative to the lower latitude sources (see Persad and Caldeira (2018), Figure 4).

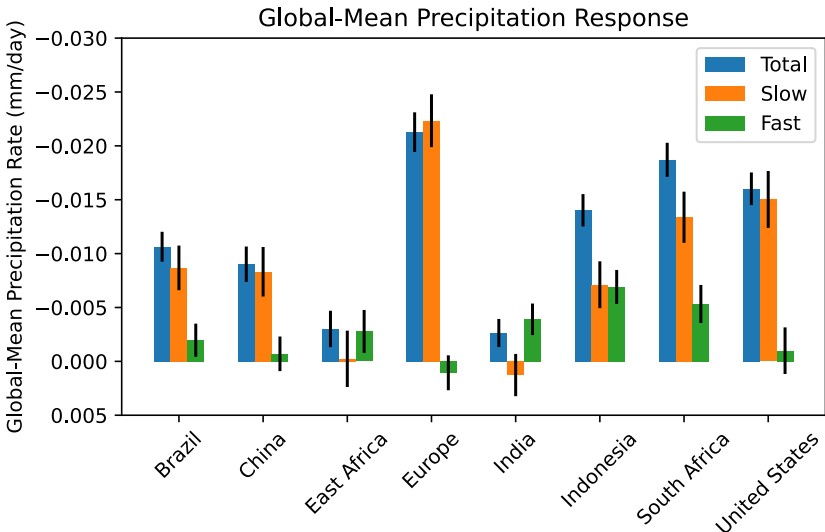

**Figure 3.** Decomposition of the global-mean total precipitation response (mm/day, blue) to identical emissions from each of the 8 regions into the slow (orange) and fast (green) precipitation response.





Emitting regions that demonstrate a substantial or dominant contribution from global-mean fast precipitation responses are those in which the aerosols produce a strong increase in global-mean atmospheric absorption in the atmosphere-only

simulations (Figure 3 and Figure 2a). Increased atmospheric absorption in the atmosphere-only simulations may result from direct radiative effects of the aerosols or from thermodynamic or fast dynamical responses in clouds and water vapor. While the amount of emissions is identical across simulations, differences in the depositional environment into which the aerosols are emitted results in varying total atmospheric concentrations in response to each regional emission (Persad & Caldeira, 2018). Emissions from India, East Africa, South Africa, and Brazil sustain the largest steady-state atmospheric burdens of black

carbon and organic carbon compared to identical emissions from the other regions (see Persad and Caldeira, 2018, Supplementary Figure 4). This partially explains the relatively high atmospheric absorption rates associated with these regional emissions and, consequently, the relatively large fast precipitation response.

**3.2 Spatial patterns of precipitation response**

The identical emissions from each region also produce differing spatial patterns of global precipitation change. A key

difference in the spatial pattern of response is a divergence in whether the regional emission generates a strong in-situ precipitation response. In all cases, precipitation decreases within the emitting region (Figure 1, Figure 4a). However, emissions from India, East Africa, Indonesia, and Brazil generate strong precipitation responses within source region boundaries, while emissions from Europe, the U.S., and China show only a minimal signature of within-region precipitation change. This signal persists when the precipitation response is normalized by the climatological precipitation (not shown),

indicating that it is not merely a function of larger climatological precipitation rates across the tropics and sub-tropics.

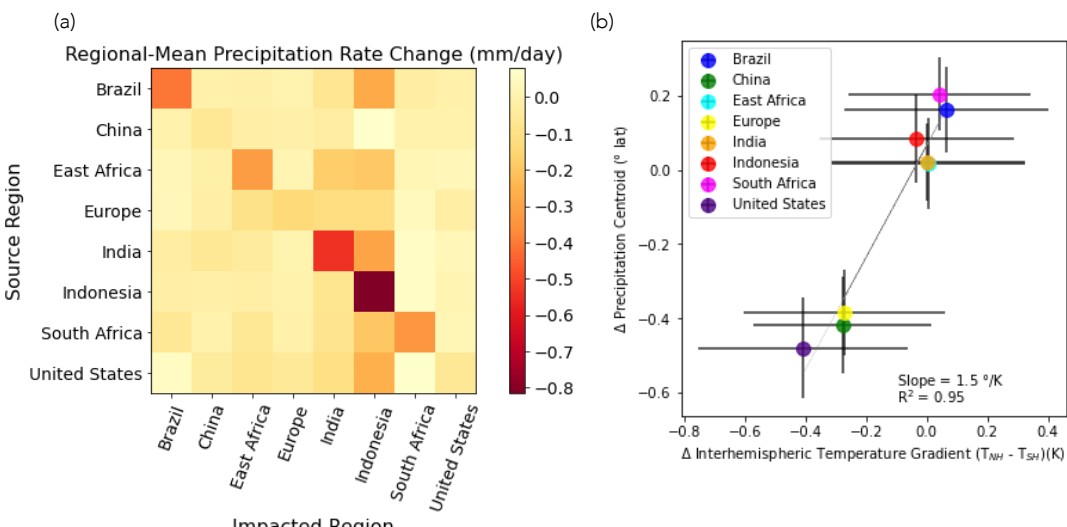

**Figure 4**. (a) Regional-mean changes in precipitation rate (mm/day) in each of the 8 regions (columns) due to emissions from each of the 8 regions (rows) are shown. (b) Shifts in the location of the intertropical convergence zone, quantified as the change in the meridional centroid of zonally averaged precipitation between 20° S and 20° N (° latitude, y-axis), correlate with the change in interhemispheric temperature

gradient, quantified as the differences between Northern Hemisphere and Southern Hemisphere mean surface temperature (K, x-axis).



Further, the maximum precipitation response does not always occur within the aerosol source region. The combination of a negligible in-situ precipitation response and strong remote precipitation response to emissions from certain regions means that aerosols from those source regions impact remote regions more strongly than themselves (Figure 1, Figure 4a). In the case of Chinese, Western European, and U.S. emissions, the maximum precipitation decline occurs remote from the source region
over the tropical oceans, whereas it occurs within the source boundaries for the other regions. In all cases, the maximum precipitation increases—which are of similar size to the maximum precipitation reductions—are dislocated from the source region and are associated with tropical precipitation shifts.

The large remote precipitation responses generated by the mid-latitude source regions result from their strong influence on the
location of the intertropical convergence zone (ITCZ), which arises from their impact on the interhemispheric temperature gradient. Figure 4b shows the change in the meridional location of the ITCZ centroid—calculated as the center of mass of the zonally averaged precipitation between 20°N and 20°S—versus the change in the interhemispheric temperature gradient— calculated as the difference between the northern and southern hemisphere mean surface temperatures—in response to emissions from each of the 8 source regions. Higher latitude sources produce larger interhemispheric temperature gradients
than lower latitude sources due to their larger total temperature effect, which manifests primarily in the source hemisphere. Northern hemisphere mid-latitude sources (Europe, the U.S., and China) produce the largest southward migration of the ITCZ, associated with their strong preferential cooling of the northern hemisphere. The southern hemisphere mid-latitude source included in these simulations (South Africa), meanwhile, generates a northward migration of the ITCZ centroid consistent with its preferential cooling of the southern hemisphere. The northern hemisphere lower latitude sources (India and Indonesia)
generate minimal (consistent with zero) shifts in either the interhemispheric temperature gradient or ITCZ centroid.

**3.3 Potential processes underlying the presence or absence of a local precipitation response**

A critical question arising from these results is why aerosol emissions from certain source regions produce a large in-situ precipitation response and others produce only a negligible one. Given the strong in-situ radiative, microphysical, and thermodynamics effects of aerosols on precipitation (e.g. Dong et al., 2019; Persad et al., 2017; Ramanathan et al., 2001), one
might expect the local response to dominate the precipitation response to regional aerosol emissions even when remote responses occur. However, these results indicate that for several source regions remote impacts may be more substantial. Aerosol emissions tend to be controlled by policy and technological decisions made at the national or subnational scale, often motivated by societally immediate and highly localized impacts on air quality (Hoesly et al., 2018; Rao et al., 2017; Riahi et al., 2017). Thus, it is important to understand whether concomitant impacts on the hydrological cycle will be similarly
concentrated or will be largely borne by others, for which source regions, and driven by what mechanisms.

One possibility investigated here is that regions with strongly localized precipitation responses are those for which the fast precipitation response contributes strongly to the total precipitation response. Because the fast precipitation response is largely





the result of in-situ stabilization and suppression of convection, evaporation, and precipitation by atmospheric absorption
(Andrews et al., 2010; Dagan et al., 2021; Samset et al., 2016), it is expected to maximize in the same regions as changes in atmospheric absorption. The changes in atmospheric absorption may result either 1) from the direct, semi-direct, and indirect radiative effects of the aerosol, which will be localized within and downwind of the emitting region due to aerosols' short atmospheric lifetime, or 2) from large-scale responses in clouds or water vapor, which are expected to be secondary to radiative effects when SSTs are fixed (Andrews et al., 2010; Samset et al., 2016). The slow precipitation response, meanwhile, results
largely from the response of the large-scale circulation and moisture transport to sea surface temperature changes, which can produce large, remote precipitation responses (Andrews et al., 2010). Thus, for emitting regions for which atmospheric absorption and fast precipitation responses are strong, but slow precipitation responses are weak, the total precipitation response should be localized to the emitting region. Conversely, emitting regions for which atmospheric absorption and the fast precipitation response are weak, but equilibrium temperature change and the slow precipitation response are strong, should
primarily generate remote precipitation responses.

This theoretical framework is borne out in the spatial patterns of response to the regional emissions imposed in this study. The fast precipitation response follows a similar spatial pattern to the changes in atmospheric absorption (Figure 5, Figure 6) and tends to be concentrated within and proximal to the emitting region. Some weaker large-scale features are evident, likely
generated by the land surface temperature changes permitted in the fixed SST simulations used to characterize the fast precipitation response (Samset et al., 2016). For all emitting regions, atmospheric absorption increases within and surrounding the emitting region—associated with the radiative effects of the combined sulfate, black carbon, and organic carbon emissions imposed—and atmospheric absorption changes remote from the emitting region are minimal. However, the strength of the atmospheric absorption response differs. The emitting regions generating the strongest localized atmospheric absorption (India,
Indonesia, East Africa, South Africa) are also those for which the fast precipitation response dominates (India, East Africa) or substantially contributes to (Indonesia, South Africa) the total precipitation response. These are also the regions in which large in-situ total precipitation responses arise (Figure 4a). Conversely, the regions that exhibit minimal in-situ precipitation responses (the U.S., Europe, and China) are also those whose emissions produce the least atmospheric absorption and the weakest fast precipitation response.








**Figure 5.** Spatial distribution of the change in fast precipitation rate (mm/day) due to the addition of an identical amount and composition of aerosol emissions in each of the 8 regions. Gridlines indicates regions where the changes are not statistically significant at the 95% confidence level via a two-sided *t* test. Global mean fast precipitation rate change and standard error in μm/day are shown in the bottom left corner of each map.





**Figure 6.** Spatial distribution of the change in atmospheric absorption (W/m$^2$) due to the addition of an identical amount and composition of aerosol emissions in each of the 8 regions. Gridlines indicates regions where the changes are not statistically significant at the 95% confidence level via a two-sided $t$ test.



## 4 Discussion

These findings highlight that understanding the processes that control the relative contribution of fast versus slow precipitation to the total precipitation response is important for constraining the magnitude and spatial distribution of precipitation response to regional aerosol forcing. In particular, understanding the processes controlling the fast precipitation response, namely atmospheric absorption, may provide an important constraint on the expected prevalence of localized precipitation responses to regional aerosol emissions. The dependence of the atmospheric absorption and fast precipitation response strength on aerosol

location seen here aligns with results from highly idealized studies. Dagan et al. (2019) forced an aquaplanet atmospheric general circulation model with equivalent, radially symmetric absorbing aerosol optical depth plumes in the deep tropics versus mid latitudes and found higher resulting atmospheric absorption in the deep tropics due to stronger cloud feedbacks. However, the aquaplanet formulation reduces the comparability of the resulting fast precipitation responses with those seen in this study. A follow-on study in the same atmosphere-only model with a realistic land surface found a stronger local fast precipitation

reduction over land in response to a tropical scattering AOD plume than to a comparable higher latitude plume, though the use of a purely scattering plume as opposed to the mixed scattering and absorbing aerosols used in this study again limits direct comparison (Dagan et al., 2021). Regardless, the results of this analysis indicate that, particularly for tropical aerosol sources, the resulting amount of atmospheric absorption—and, consequently, the strength of the fast precipitation response—can be a strong determinant of the overall precipitation response. The importance of the atmospheric absorption for the precipitation

response is particularly notable, since scattering rather than absorption by the mixed aerosol emissions in this study dominates the temperature and overall radiative response (Figure 2b and Persad and Caldeira (2018)). There are known model biases and limited observational constraints on atmospheric absorption, particularly at the regional scale (Samset et al., 2018). The importance to the hydrological response seen here, however, reinforces the need for improved observations and modeling of the processes that control atmospheric absorption.


The regions that manifest local fast versus slow precipitation responses in the simulations analyzed here also overlap with regions identified in existing studies. Samset et al. (2018) evaluated the regions for which fast precipitation responses dominate slow precipitation responses for multi-model simulations of idealized global forcings, including 10 times present-day global black carbon emissions and 5 times present-day global sulphate emissions. Although the spatial pattern of imposed perturbation

differs from this study (i.e. globally distributed vs. regionally confined perturbations), they also find that the total precipitation response to both global BC and global sulphate are dominated by the fast response over parts of South Asia and most of the African continent. High latitude precipitation responses to these two forcers, meanwhile, are dominated by the slow precipitation response in the multi-model simulations (Samset et al., 2016), though individual models show conflicting results (Zhang et al., 2021). Similar multi-model simulations with regional idealized aerosol emissions over Asia and Europe (Liu et

al., 2018), however, also showed a strong local fast precipitation response to Asian aerosols and almost no fast precipitation





response to European aerosols. The appearance of a fast precipitation response in low latitude continental regions in response to both localized and global-scale aerosol forcing and the absence of one at high latitudes thus appears to be a robust feature across models and aerosol perturbation set-ups.

The latitudinal dependence of the strength of the fast precipitation response and consequently the total local precipitation response to a regional aerosol emission may bear the signature of other processes that differentiate between the tropics and extratropics. For example, differences in local energy budget closure and the strength of horizontal energy and moisture gradients between the tropics and extratropics, associated with poleward strengthening of the Coriolis force, have been leveraged to explain the latitudinal dependence of fast precipitation responses to idealized aerosol forcing (Dagan et al., 2019,

2021). The relative role of local- versus large-scale precipitation processes in supplying precipitation to a region may also play a role in its susceptibility to in-situ aerosol forcing. The tropics and extratropics differ strongly in the proportion of total precipitation that is supplied by convective versus large-scale precipitation (Figure A2), though specific patterns are highly model dependent (Dai, 2006; Kyselý et al., 2016). To the (imperfect) extent to which these two model-derived flavors of precipitation correspond with precipitation that is controlled by local- versus large-scale processes in the real world (Norris et

al., 2021), regions with climatological precipitation dominated by convective (i.e. local-scale) precipitation may be more susceptible to local aerosol forcing. Large-scale precipitation processes, meanwhile, may be relatively insensitive to localized forcing from a regional aerosol emission, since they are more strongly controlled by large-scale moisture and energy gradients (Wang et al., 2021).

The plausibility of this process is hinted at by the fact that regions that exhibit strong in-situ precipitation responses to local aerosol emissions (Figure 1, Figure 4a) are also those whose climatological precipitation is overwhelmingly supplied by convective precipitation (Figure A2). Conversely, the regions that show a negligible local response are those in which climatological precipitation is partly or primarily supplied by large-scale precipitation. Additionally, the local precipitation response is dominated by convective precipitation change in all cases (Figure A3). However, it should be noted that CAM5,

like many CMIP5 and CMIP6 generation models, includes aerosol microphysical effects on precipitation in its convective precipitation scheme, but not its large-scale precipitation scheme (Hurrell et al., 2013). Thus, this signal may be largely dependent on parameterization approach. Applying well-established energetic analysis approaches (Dagan et al., 2021; Liu et al., 2018; Ming et al., 2010; Zhang et al., 2021) to more realistic aerosol perturbations in combination with moisture tracking algorithms (Mei et al., 2015) that allow better characterization of precipitation sources could help further clarify the potential

role of local versus large-scale precipitation controls in determining the emergence of in-situ precipitation responses to regional aerosols.

The greater capability of higher latitude emission sources at generating total global-mean precipitation change also appears to be a robust feature of the response to aerosols. Studies analysing the global-mean precipitation response to removal of present-



day aerosols from individual regions find that removal of European or North American emissions generates stronger global-mean precipitation per unit of radiative forcing than removal of South or East Asian emissions (Kasoar et al., 2018; Westervelt et al., 2020), in line with the findings here. This reinforces the latitudinal dependence in the climate response to heterogeneous regional forcing found in earlier studies (Shindell & Faluvegi, 2009; Shindell et al., 2012) and indicates that it continues to apply as the forcings become more regionalized.


Despite the promising alignment of this study's findings with prior single and multi-model work, the single-model, slab-ocean set-up used here for computational tractability may create biases that encourage future multi-model coupled investigation of these questions. The regional aerosol response in the fully coupled NCAR CESM1 model is comparable with results from other contemporary coupled models, such as GFDL CM3, but the ITCZ response to aerosols is somewhat stronger (Westervelt

et al., 2018). Ocean dynamical adjustments present in the fully coupled model but not in the slab ocean configuration used here can either damp or amplify the response to aerosols, depending on the spatial pattern of forcing (Kang et al., 2021). In particular, they may damp ITCZ shifts relative to those seen in a slab ocean model (Zhao & Suzuki, 2019). Nevertheless, slab ocean models can provide valuable insights on hydrological cycle responses to anthropogenic forcings when computational efficiency is needed (Held & Soden, 2006; Ming & Ramaswamy, 2009). The strong dependence of the hydrological response

to aerosols on regional distribution seen here and in other studies, across perturbation set-ups and models, highlights the importance of continued investment in developing a comprehensive and consensus theory of what drives this dependence.

**5 Conclusions**

This study identifies a strong dependence of the global-mean precipitation response and its spatial distribution on the geographic location of a given aerosol emission. Coupled atmosphere-slab ocean GCM simulations, in which an identical,

quasi-realistic amount and composition of aerosol is separately placed into a range of emitting regions, are used to isolate the importance of source location in determining aerosols' hydrological effects. A six-fold difference in global-mean precipitation response emerges, largely driven by a fourteen-fold difference in global-mean temperature response. This arises from a combination of differing strengths of global-mean radiative forcing generated by each regional emissions and diversity in the efficacy of that radiative forcing at generating cloud and ice albedo feedbacks (Persad & Caldeira, 2018). Major distinctions

in the geographic distribution of the precipitation response, particularly the prevalence of local versus remote responses, also arise. Tropical regions, which also tend to have the greatest societal vulnerability to precipitation disruption, are the most susceptible to dynamically-driven precipitation changes generated by both local and remote aerosol emissions, as has been highlighted elsewhere (e.g. Scannell et al., 2019; Westervelt et al., 2018; Zanis et al., 2020). However, these regions are also the least effective at generating remote precipitation responses. Conversely, mid-latitude emissions sources are highly effective

at generating global-mean precipitation change, but these changes occur almost entirely outside of the source region. These results indicate that the time-evolving geographic distribution of global aerosol emissions may have substantial implications





for the precipitation stability of vulnerable regions outside of key past or projected emissions hotspots. Building a comprehensive theory of what determines the relative strength of local versus remote hydrological impacts of regional aerosol emission changes will be important to understanding and predicting these trends.


The continuous spatial redistribution of aerosol emissions constitutes an ongoing source of uncertainty and variability in the global hydrological cycle (Deser et al., 2020), and near-term regional aerosol trends may be a major determinant of climate risk over the next several decades (Luo et al., 2020; Samset et al., 2019). Notably, the regions identified in this study to have the strongest global-mean and remote precipitation impacts (Europe and the U.S.) have seen strongly declining aerosol emission since the mid-20th century (Hoesly et al., 2018). Conversely, those with the weakest global-mean but the strongest local precipitation impacts (e.g. India, Indonesia, East Africa) are among those in which emissions could continue to increase substantially through the mid-21st century (Lund et al., 2019). This implies that the future spatial distribution of aerosol emissions may have a lower overall effectiveness at changing global-mean precipitation but may preferentially concentrate precipitation impacts into vulnerable regions. Known nonlinearity in the climate response to simultaneous regional aerosol variations (Herbert et al., 2021) limits the direct application of the regional dependence quantified here to estimate how ongoing spatial redistribution of aerosol emissions will affect global precipitation patterns. However, the strong dependence on source location seen in this study demonstrates that the geographic distribution of aerosol emissions must be accounted for when quantifying the human influence on global precipitation patterns.





## Appendix A: Additional Figures

**Figure A1**. An identical total change in emissions is distributed within each of the 8 perturbation regions according to patterns shown above. Distributions are shown in terms of the percent of the identical total emissions change that occurs within a given model grid cell within the region and follows the year 2000 realistic distribution of emissions within that region.

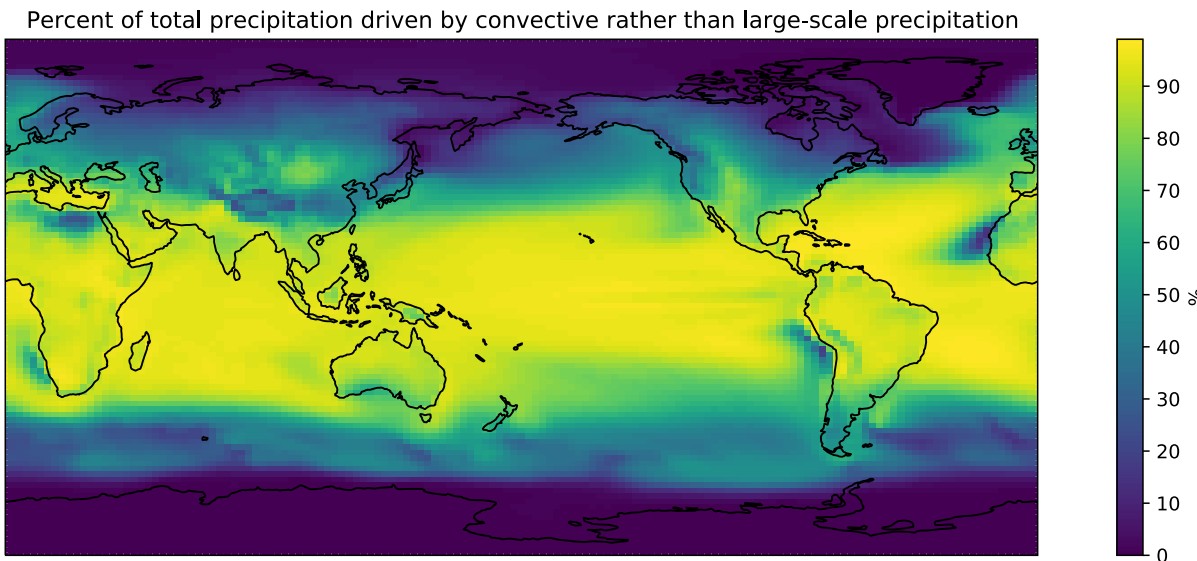

**Figure A2.** The percent of total precipitation in each grid cell in the control simulation that is derived from convective precipitation as opposed to large-scale precipitation is shown.

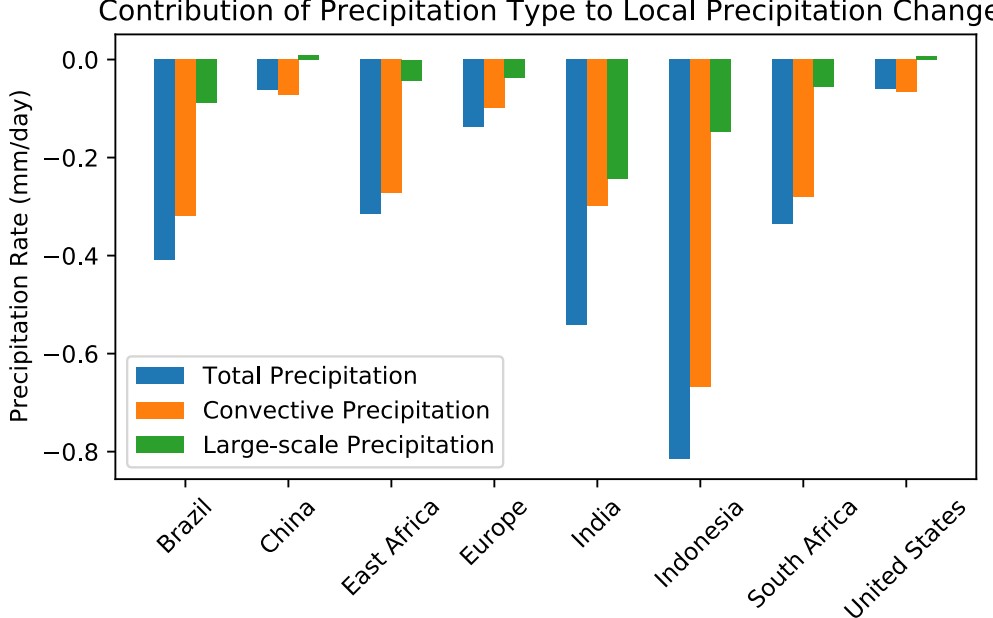

**Figure A3.** The relative contribution of convective precipitation change (mm/day, orange) and large-scale precipitation change (mm/day, green) to the total local precipitation response (mm/day, blue) within each of the 8 perturbation regions in response to emissions within that region are shown.




**Code availability**

The NCAR CESM1.2 model is an open source model and is publicly available at https://www.cesm.ucar.edu/models/cesm1.2/.

**Data availability**

Input emissions data files developed for the perturbation simulations are citeable and available for download via the Texas
Data Repository at https://doi.org/10.18738/T8/Z87COZ. All output data analysed as part of this study are also citeable and available for download via the Texas Data Repository at https://doi.org/10.18738/T8/WBNQZE. All other input data used are available as part of the standard public release of the NCAR CESM1.2 model (see Code availability).

**Author contribution**

G. G. Persad conceived of the study, conducted the simulations and analysis, and wrote the paper.

**Competing interests**

The author declares no competing interests.

**Acknowledgements**

This material is based upon work supported by the National Science Foundation under Grant No. 1715557.



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
