# Peer review of "The Dependence of Aerosols' Global and Local Precipitation Impacts on Emitting Region"

_Atmospheric Chemistry and Physics, 2022_

## Author Comment (AC1)

In this paper the author uses a series of single model simulations with variations in the aerosol emission's geographical location to study the dependence of precipitation on aerosol emission location. The manuscript is well written, nicely organized and study interesting and important topic. However, I have a strong concern that many of the results are not significantly differ from changes one can get just due to internal variability. In the global mean, the changes presented here are in the range of ~ 0.002-0.02 mm/day. I have calculated the pre-industrial (PI) CESM1 global mean precipitation to be 3.044 mm/day, i.e., the changes seen here are in the range of 0.07-0.7% of the global mean. A question which is central to this paper is whether the changes reported here are statistically significant compared to the natural verbality of the system. The range of the 40-year running mean global mean precipitation in the CESM1 PI is 3.037 -3.051 mm/day, i.e., a range of 0.014 mm/day just due to natural viability. That means that at least some of the difference in the global mean precipitation seen here are within the range possible only due to natural variability (all thought some of them are outside this range). Looking on the significant test in Fig. 1, suggests that the same might be true for the local precipitation changes (the changes in the vast majority of places aren't significant). In addition, statistically significant local precipitation variations between two realizations could also be driven by natural variability and not by the external (aerosol) forcing.

The way to overcome this issue, and to make sure that the differences are driven by the aerosol forcing and not by natural variability is to simulate more realizations (i.e., conduct initial-condition large-ensemble) (Diao et al., 2021), or run the model for very long times (Fiedler & Putrasahan, 2021). I feel bad to ask such a revision as I understand the amount of work it might require. However, I hope that, if the author will accept my suggestion, the paper could become much more convincing. If conducting a large-ensemble is beyond the reach of the author, I believe that conducting one or two more simulations for each aerosol location (with slightly different initial conditions) will improve the confidence in the results (in case they are similar to the initial results) or demonstrate the need in more realizations (in case they are not similar, thus suggesting a large role of natural variability).

I thank the referee for their thoughtful consideration of the results and for their favorable review of the overall manuscript structure and content. I agree that characterizing the statistically significant precipitation response to aerosol perturbations is a difficult process. However, I believe the referee's concerns regarding statistical significance are based on a misapprehension. I detail this below, as well as steps that are taken in the revised manuscript to make the reliability of the signals clearer. I nevertheless appreciate the interrogation of the statistical significance calculation, as this is an important factor to clarify. The PI CESM1 variability the referee estimates appears to be from the fully coupled CESM1, which will have different variability than the slab-ocean set-up used in this study (see Section 2). Measures of statistical significance should be derived from the native dataset. I use 60 years of simulations in the slab-ocean configuration to characterize the precipitation signal. Standard errors derived from the interannual variability in the difference between the slab ocean PI control and each perturbation experiment were previously given in Figure 1. The standard error values may be roughly doubled to provide an estimate of the 95% confidence range. This 95% confidence range is also provided as error bars on the fast and slow precipitation responses in Figures 2 and 3 (note that this information has now been added to the relevant figure caption). From this significance calculation, it is clear that for all regional perturbations aside from Indian and East African emissions the global-mean total precipitation response is statistically distinguishable from zero at the 95% confidence level after accounting for internal variability and thus is highly unlikely to have arisen from internal variability alone. (The global-mean precipitation response to East African and Indian emissions are statistically indistinguishable from each other and from zero.)

Furthermore, the paper does not claim that every global-mean response is statistically distinct from the others, nor does its argumentation depend on this. There is clearly statistically significant diversity in the global-mean response to identical aerosol emissions from different regions (a range of -2.6 to -21.3  $\mu$ m/day compared to a maximum standard error of 1.8  $\mu$ m/day and a maximum 95% confidence interval of 3.6  $\mu$ m/day), which forms one of the motivations for the analysis in the paper.

Regarding the statistical significance shown on the maps, it is to be expected that regional aerosol perturbations will produce spatially heterogeneous precipitation changes that will be statistically significant in some regions and not in others. The spatial extent of regions with statistical significance is comparable or higher than that seen in other modeling studies of regional aerosol effects on precipitation using forcings of similar magnitude across several different climate models and a similar statistical significance criterion. For examples, see the following:

- Westervelt et al., 2017, Figure 3 fully coupled 200-year simulations of removal of U.S. sulfate emissions in NCAR CESM, GFDL CM3, and GISS E2 models
- Westervelt et al., 2018, Figure 1 fully coupled 200-year simulations of removal of U.S., European, and Asian sulfate, bc, or all anthropogenic aerosol emissions in NCAR CESM, GFDL CM3, and GISS E2 models
- Kasoar et al., 20218, Figure S2 fully coupled 150-year simulations of removal of North American, European, East Asian, or South Asian sulfate emissions in HadGEM3-GA4

Westervelt, D. M., Conley, A. J., Fiore, A. M., Lamarque, J.-F., Shindell, D., Previdi, M., Faluvegi, G., Correa, G., & Horowitz, L. W. (2017). Multimodel precipitation responses to removal of U.S. sulfur dioxide emissions. *Journal of Geophysical Research: Atmospheres*, *122*(9), 2017JD026756. https://doi.org/10.1002/2017JD026756.

Westervelt, D. M., Conley, A. J., Fiore, A. M., Lamarque, J.-F., Shindell, D. T., Previdi, M., Mascioli, N. R., Faluvegi, G., Correa, G., & Horowitz, L. W. (2018). Connecting regional aerosol emissions reductions to local and remote precipitation responses. *Atmospheric Chemistry and Physics*, *18*(16), 12461–12475. https://doi.org/10.5194/acp-18-12461-2018

Kasoar, M., Shawki, D., & Voulgarakis, A. (2018). Similar spatial patterns of global climate response to aerosols from different regions. *Npj Climate and Atmospheric Science*, *1*(1), 12. https://doi.org/10.1038/s41612-018-0022-z

The following improvements will be made to the revised manuscript:

- I now report the 95% confidence interval rather than the standard error on Figure 1 and wherever the global-mean total precipitation responses are reported in the text. Error ranges given on Figure 5 (fast precipitation response) and Figure 3 are now also the 95% confidence interval rather than the standard error as well. All error ranges throughout the manuscript thus now provide the 95% confidence interval, allowing clear depiction of whether all signals are is distinguishable from natural variability with high confidence.
- I summarize the above discussion of statistical distinctness of the global-mean precipitation response as follows (L139-145):
  "The global-mean precipitation response to Indian and East African emissions, which constitute the weakest of the precipitation responses, are statistically indistinguishable from zero and from each other in the presence of internal variability. All other global-mean precipitation responses are statistically significant at the 95% confidence level, and thus highly unlikely to arise from internal variability alone. Although the 95% confidence interval in the global-mean response to some regional emissions are overlapping, it is clear that there is statistically significant diversity in the global-mean response to identical aerosol emissions from different regions."
- I now explicitly indicate on Fig. 4a whether the regional-mean precipitation responses to in-situ aerosol changes are statistically significant at the 95% confidence level.
- I now improve the density and visibility of the statistical significance masking on all map figures in the main text (Figures 1, 5, and 6).

In addition, I believe that presenting Fig. 4 in relative terms will be more appropriate as the difference in the background precipitation between these places is very large.

I now include the precipitation response as a percent of the climatological precipitation in each grid box in the supplementary materials (I also include it below). As already stated in the manuscript, the distinction between regional emissions producing strong versus weak in-situ precipitation responses holds true whether the precipitation responses are considered in terms of absolute values or percent values. However, I have chosen to keep the absolute precipitation change as the format for the precipitation maps in the main manuscript, as this allows a more accurate portrayal of the spatial pattern of precipitation change (percent values may amplify the appearance of the precipitation response in some regions, if the denominator is small). It is for this reason that absolute precipitation changes are the standard mode of depiction of spatial patterns of precipitation response across recent papers exploring precipitation responses to regional aerosols, e.g. Westervelt et al. (2017), Westervelt et al. (2018), and Kasoar et al. (2018) cited previously, as well as Liu et al. (2018) and Samset et al. (2016).

Liu, L., Shawki, D., Voulgarakis, A., Kasoar, M., Samset, B. H., Myhre, G., Forster, P. M., Hodnebrog, Ø., Sillmann, J., Aalbergsjø, S. G., Boucher, O., Faluvegi, G., Iversen, T., Kirkevåg, A., Lamarque, J.-F., Olivié, D., Richardson, T., Shindell, D., & Takemura, T. (2018). A PDRMIP Multimodel Study on the Impacts of Regional Aerosol Forcings on Global and Regional Precipitation. *Journal of Climate*, *31*(11), 4429–4447. https://doi.org/10.1175/JCLI-D-17-0439.1

Samset, B. H., Myhre, G., Forster, P. M., Hodnebrog, Ø., Andrews, T., Faluvegi, G., Fläschner, D., Kasoar, M., Kharin, V., Kirkevåg, A., Lamarque, J.-F., Olivié, D., Richardson, T., Shindell, D., Shine, K. P., Takemura, T., & Voulgarakis, A. (2016). Fast and slow precipitation responses to individual climate forcers: A PDRMIP multimodel study. *Geophysical Research Letters*, *43*(6), 2016GL068064. https://doi.org/10.1002/2016GL068064

---

## Author Comment (AC2)

This paper by Geeta G. Persad shows how precipitation response depends on aerosol emission regions. The paper is well written and the topic is relevant for the community.

Author use CESM2-CAM5 model with slab ocean configuration. Experimental setup consists of 8 regions where the author has changed regional emissions to correspond to China's emissions from the year 2000. Author clearly shows how the fast and slow precipitation responses depend on the emissions regions, and discusses thoroughly on the mechanisms behind the changes.

I thank the reviewer for their careful assessment of the paper and their overall favorable review of its clarity and relevance to the community.

Major comments

The role of natural variability is not discussed. As the runs are equilibrium runs, the year-to-year variability can be used as an estimate for natural variability. Are the results significant compared to year-to-year variability.

The focus of the paper is on the forced signal, thus extensive discussion of the role of natural variability is outside the scope. However, the statistical significance estimates provided on all figures assesses the significance of the forced signal compared to year-to-year variability. I have now enhanced the presentation of the statistical significance estimates and added explicit discussion of their relation to year-to-year variability.

I use 60 years of simulations in the slab-ocean configuration to characterize the equilibrium precipitation response to aerosol perturbations and use the year-to-year variability as an estimate of natural variability. Standard errors derived from the interannual variability in the difference between the slab ocean PI control and each perturbation experiment were previously given in Figure 1 and have now been updated to provide the 95% confidence range. This 95% confidence range is also provided as error bars on the fast and slow precipitation responses in Figure 2. From this significance calculation, it is clear that for all regional perturbations aside from Indian and East African emissions the global-mean total precipitation response is statistically distinguishable from zero at the 95% confidence level after accounting for internal variability and thus is highly unlikely to have arisen from internal variability alone. Similar statistical significance estimates have been provided for all other figures.

The following improvements will be made to the revised manuscript, including explicit discussion of how the statistical significance calculations reflect the role of year-to-year variability:

- I now more explicitly discuss the interpretation of the statistical significance calculation in the context of internal year-to-year variability as follows:
  - L120-122 (Methods): "The 95% confidence level (i.e. 1.96$\sigma$) based on year-to-year variability in the difference between the control simulation and each perturbation experiment is provided for all global-mean values."
  - L139-145 (Results): "The global-mean precipitation response to Indian and East African emissions, which constitute the weakest of the precipitation responses, are statistically indistinguishable from zero and from each other in the presence of internal variability. All other global-mean precipitation responses are statistically significant at the 95% confidence level, and thus highly unlikely to arise from internal variability alone. Although the 95% confidence interval in the global-mean response to some regional emissions are overlapping, it is clear that there is statistically significant diversity in the global-mean response to identical aerosol emissions from different regions."
- The above information on statistical significance was previously omitted accidentally from the figure captions, as the reviewer has noted in the minor comments. I have now updated the figure captions in Figures 2, 3, and 4 to make clearer that the 95% confidence interval is indicated.
- All error ranges or statistical significance markings shown on all figures now use the 95% confidence interval (instead of the standard error) to provide explicit indication of whether the signal can be confidently distinguished from year-to-year variability.

How do these results compare to other similar experiments with other models? Example PDRMIP regional experiments

Much of the Discussion section is devoted to comparing the results of this paper to similar experiments in other models, including the PDRMIP regional experiments analyzed in Liu et al., 2018. In general, the fundamental physical behavior exhibited by these simulations is well-aligned with the results of other similar experiments. I now more explicitly indicate the experimental design and models used in the studies that are compared with throughout the discussion section as follows:

L286-294: "The dependence of the atmospheric absorption and fast precipitation response strength on aerosol location seen here aligns with results from highly idealized studies. Dagan et al. (2019) forced an aquaplanet atmospheric general circulation model (ICON) with equivalent, radially symmetric absorbing aerosol optical depth plumes in the deep tropics versus mid latitudes and found higher resulting atmospheric absorption in the deep tropics due to stronger cloud feedbacks. However, the aquaplanet formulation reduces the comparability of the resulting fast

precipitation responses with those seen in this study. A follow-on study in the same atmosphere-only model with a realistic land surface found a stronger local fast precipitation reduction over land in response to a tropical scattering AOD plume than to a comparable higher latitude plume, though the use of a purely scattering plume as opposed to the mixed scattering and absorbing aerosols used in this study again limits direct comparison (Dagan et al., 2021)."

L303-316: "The regions that manifest local fast versus slow precipitation responses in the simulations analyzed here also overlap with regions identified in existing studies, including those utilizing Precipitation Driver and Response Model Intercomparison Project simulations (PDRMIP, Myhre et al., 2016). Samset et al. (2018) evaluated the regions for which fast precipitation responses dominate slow precipitation responses for PDRMIP multi-model simulations of idealized global forcings, including 10 times present-day global black carbon emissions and 5 times present-day global sulphate emissions. Although the spatial pattern of imposed perturbation differs from this study (i.e. globally distributed vs. regionally confined perturbations), they also find that the total precipitation response to both global BC and global sulphate are dominated by the fast response over parts of South Asia and most of the African continent. High latitude precipitation responses to these two forcers, meanwhile, are dominated by the slow precipitation response in the multi-model simulations (Samset et al., 2016), though individual models show conflicting results (Zhang et al., 2021). Similar PDRMIP multi-model simulations with regional idealized aerosol emissions over Asia and Europe (Liu et al., 2018), however, also showed a strong local fast precipitation response to Asian aerosols and almost no fast precipitation response to European aerosols. The appearance of a fast precipitation response in low latitude continental regions in response to both localized and global-scale aerosol forcing and the absence of one at high latitudes thus appears to be a robust feature across models and aerosol perturbation set-ups. "

L346-353: "The greater capability of higher latitude emission sources at generating total global-mean precipitation change also appears to be a robust feature of the response to aerosols. Studies analysing the global-mean precipitation response to removal of present-day aerosols from individual regions find that removal of European or North American emissions generates stronger global-mean precipitation per unit of radiative forcing than removal of South or East Asian emissions (via fully coupled HadGEM3-GA4 simulations in Kasoar et al., 2018; via fully coupled GISS E2, GFDL CM3, and NCAR CESM1 simulations in Westervelt et al., 2018), in line with the findings here. This reinforces the latitudinal dependence in the climate response to heterogeneous regional forcing found in earlier studies (Shindell & Faluvegi, 2009; Shindell et al., 2012) and indicates that it continues to apply as the forcings become more regionalized."

minor comments:

Figure 2. Lack of explanation for the black lines

Thank you for catching this omission. All black lines are the error bars associated with the 95% confidence interval. The caption for the figure has been updated to describe this as follows:

"Error bars provide the 95% confidence interval ($\pm 1.96\sigma$)."

Figure 1,figure 6,figure 5. It is somewhat hard to read where the precipitation change is significant when the statistical significance is indicated via gridlines. Meaby change to dots?

The density of gridlines has been increased to make the regions of statistical significance more clearly distinguishable. A different output format has also been used to reduce issues with rendering when saving to PDF, which altered the visibility of the gridlines.

Figure 3. Lack of explanation for the black lines

Thank you for catching this omission. These black lines are the error bars associated with the 95% confidence interval. Note that this has been updated from the previous version of this figure, which used standard error rather than 95% confidence interval (see response to Major Comments above). The caption for the figure has been updated to describe this as follows:

"Error bars provide the 95% confidence interval ($\pm 1.96\sigma$)."

Lines 150-155. Here author list different feedbacks due to sea surface changes. I would like to see also how this is limited by the slab ocean configuration

I do expect that some aspects of the responses seen here would be modified by use of a fully dynamical rather than slab ocean, as is discussed in detail at L355-365. In the context of the slow precipitation scaling and associated driving processes discussed at L150-155, however, it is important to note that one of the seminal papers to identify this 2-3%/K precipitation increase per degree K and the associated driving moist convective feedbacks and dynamical constraints was conducted in a slab ocean configuration like the one use here (Held and Soden, 2006). Although that analysis was done on the total rather than slow precipitation response, it was done in the context of

the response to $CO_2$ forcing for which the fast precipitation response is expected to be overwhelmed by the slow precipitation response. This behavior has since been confirmed in response to a broader range of forcings and using dynamical ocean set-ups in Samset et al., 2016 and Sillman et al., 2017. Notably, the slab ocean simulations used here also exhibit this 2-3%/K scaling in the slow precipitation response to regional aerosol emissions (identifiable from Figure 2b). Unfortunately, explicit assessment of whether the same tropical mass flux constraint identified by Held and Soden (2006) is operating here would require analysis of the convective mass flux, which was not saved out in these simulations.

I have added the below language at L165-167 to summarize the above discussion:

"Indeed, the slow precipitation response to regional aerosol perturbations seen here follows the 2-3%/K scaling previously identified in both fully dynamical ocean and slab ocean coupled set-ups (Held & Soden, 2006; Samset et al., 2016; Sillmann et al., 2017)."

line 115-120,145-146: Change word couple to slab ocean, to indicate that runs are not done with fully coupled ocean.

These modifications have been made. Note that I have replaced "coupled" with "slab ocean coupled" at L152 and 153 to distinguish from the atmosphere-only results, as these responses do involve atmosphere-ocean coupling (just to a slab rather than fully dynamical ocean).

Figure 4a. should show also if the change is significant or not, example by hatching the squares.

Thank you for this suggestion. I have now updated Figure 4a to indicate the regional-mean responses that are statistically significant at the 95% confidence level (black asterisks). However, it should be noted that many of the regions that do not exhibit a statistically significant regional-mean precipitation response do exhibit statistically significant precipitation responses in some grid cells. I have therefore also distinguished the regional-mean responses that do not have any statistically significant within-region precipitation responses (grey asterisks) from those that do (no asterisk).

[Figure]

**Figure 4**. (a) Regional-mean changes in precipitation rate (mm/day) in each of the 8 regions (columns) due to emissions from each of the 8 regions (rows) are shown. (b) Shifts in the location of the intertropical convergence zone, quantified as the change in the meridional centroid of zonally averaged precipitation between 20° S and 20° N (° latitude, y-axis), correlate with the change in interhemispheric temperature gradient, quantified as the differences between Northern Hemisphere and Southern Hemisphere mean surface temperature (K, x-axis). Error bars on panel (b) provide the 95% confidence interval (±1.96σ). Black asterisks on panel (a) indicate regional-mean precipitation changes that are significantly different than zero with 95% confidence. Grey asterisks on panel (a) indicate regions with no statistically significant precipitation response in any grid cell; All others show statistically significant precipitation responses in some grid boxes within the region (see Figure 1), although the regional-mean change is not statistically significant.

---

## Referee Report (RR1)

This is my second time viewing this paper and all my precius comments are satisfactory addressed.

---

## Author Response (AR2)

**Reviewer 1**

I would like to thank the author for replying to my comments.
From the reply it seems like, maybe, I was not clear enough about my main concern. If that was the case, I am sorry.

The author demonstrates, using the standard error of 60 years slab ocean simulations, that (at least in some of the cases) simulations with added anthropogenic aerosols at a given location are different from a reference simulation, which include no anthropogenic aerosols. I have no concern about this part. My concern is about the attribution of the difference to the added aerosols. Two different 60 year-long simulations could by statistical different from each other (in a 95% confidence level) even without any external forcing, i.e., only due to internal variability. Based on the presented results in this paper, one can't rule out that this is the case here. This simply cannot be done in a relatively short (60 years) single simulation for each forcing pattern (unless the signal is orders of magnitude larger than the range possible due to internal variability, which is not the case here, even compared to slab ocean simulations and not fully coupled simulations).
In order to robustly attribute precipitation changes to aerosols, large ensemble of simulations or very long (1000's years) simulations are needed. Hence, I would still like to encourage the author to conduct, at the very least, one or two more simulations for each aerosol location (with slightly different initial conditions). This could at least strength the attribution argument, even though without a large ensemble (few 10's) of simulations it will not completely rule-out the role of internal variability.
Finally, a word about the comparison of slab ocean and fully-coupled ocean simulations. No doubt that the estimation of the statistical significance, as well as of the role of the forced response (compared to natural variability) should be derived from a similar dataset. However, the atmospheric natural variability is still very high in slab ocean simulations and the range of precipitation due to natural variability alone is not expected to be much smaller than in coupled simulation. In fact, even in simulations with prescribed SST a large-ensemble is needed many times to identify atmospheric response (see for example (Gervais et al., 2019)).

Gervais, M., Shaman, J., and Kushnir, Y.: Impacts of the North Atlantic warming hole in future climate projections: Mean atmospheric circulation and the North Atlantic jet, Journal of Climate, 32, 2673-2689, 2019.

I thank the reviewer for further clarifying their concerns regarding the methodology used in this study. Despite the fact that the methodology and significance testing applied in this study to identify climate responses attributable to changes in regional aerosol emissions is standard across the aerosol-climate literature (see below), I appreciate the opportunity to advance best practices and have made several improvements to the manuscript to alleviate any concerns regarding the robustness of the signals assessed.

- All slab ocean coupled simulations have been repeated with slightly adjusted initial conditions. All analysis conducted on the original simulation set is now repeated in this second simulation set, and comparable figures for the second simulation set are now included in Appendix A. The results from the second simulation set are consistent with the original simulation set, indicating that the signal is unlikely to arise from internal variability.

- The concern raised by the reviewer that "two different 60 year-long simulations could by statistical different from each other (in a 95% confidence level) even without any external forcing, i.e., only due to internal variability" could emerge if a persistent mode of internal variability was present in the perturbation simulation and thereby conflated with the perturbation signal. This is addressed by the additional simulations described

above. However, it was also addressed in the original submission by adjusting the effective sample size of all statistical tests for autocorrelation between years. Any persistent mode of internal variability would be expected to increase autocorrelation between years. Were a persistent mode to produce the signals evaluated, it would be expected to reduce the effective sample size to the point where significant signals would not be detected. Following best practices (e.g. Westervelt et al. 2020, Conley et al., 2018), I adjust the effective sample size to account for autocorrelation following the methodology of Santer et al. (2000). This practice is now described more clearly in the Methodology section of the manuscript.

- I would like to note that I use the 95% confidence interval to characterize uncertainty throughout, not the standard error as the reviewer initial stated.

Results of the new simulation set are provided in Figures A2-A5 and the above improvements to the manuscript are detailed in Section 2, L119-144 of the revised manuscript as follows:

"The simulation design used here, in which a signal from a given perturbation (e.g. a regional aerosol emissions) is characterized by imposing that perturbation as the only modification to a control simulation and running the resulting simulation in repeating annual cycle mode for an extended period, is a standard methodology used across the aerosol-climate interactions literature. Examples include simulations conducted as part of the Precipitation Driver and Response Model Intercomparison Project (PDRMIP) (e.g. L. Liu et al., 2018; Myhre et al., 2016; Samset et al., 2016) with idealized regional aerosol perturbations and within multi-model (Westervelt et al., 2017, 2018) and single-model experiment designs (Kasoar et al., 2018) simulating removal of present-day aerosol emissions in individual regions. In this experiment design, the perturbation signal is characterized as the difference between the long-term mean of the perturbation and control experiments after they have reached quasi-equilibrium, and the effects of internal variability are estimated using the interannual variability between individual years of the simulation.

One concern with this approach, not addressed in prior studies, is that persistent modes of internal variability may emerge within the equilibrium simulations and could be conflated with the perturbation signal. While atmosphere-only simulations cannot sustain long-term modes of internal variability, this concern may apply to the slab ocean coupled simulations used here. To address this concern, two approaches are applied in this study. First, statistical significance is estimated using either the last 60 years (slab ocean simulations) or 40 years (atmosphere-only) of the simulations as the sample, but effective sample size is adjusted to account for autocorrelation between simulation years following the methodology of (Santer et al., 2000). The 95% confidence level (i.e. $1.96\sigma$) based on year-to-year variability in the difference between the control simulation and each perturbation experiment is provided for all global-mean values and statistical significance for maps is estimated at the 95% confidence level using a two-tailed $t$ test, both using this adjusted effective sample size. Second, the slab ocean coupled experiments are repeated with slightly adjusted initial conditions (initial conditions drawn from a different year of the control simulation), allowing a different trajectory of internal variability to emerge within the equilibrium simulation. Results from this second experiment set are provided in Appendix A and demonstrate that the central findings of the study are unlikely to be the result of persistent modes of internal variability emerging in either equilibrium

simulation set, but rather can robustly be assumed to result from the regional aerosol emissions perturbations imposed."

I would, however, like to highlight that the methodology and significance testing applied in this study to identify climate responses attributable to changes in regional aerosol emissions is standard across the aerosol-climate literature. All of the below studies are conducted as repeating annual cycle equilibrium simulations, as in the current study. Several studies use perturbations comparable in magnitude to that imposed in this study (Westervelt et al., 2020, Westervelt et al., 2018, Kasoar et al., 2016, Kasoar et al., 2018) and run for a similar (O(100) years)) duration. Notably, none of them run multiple ensemble members for a given equilibrium simulation, as the purpose of the equilibrium simulation is to sample internal variability across the duration of the simulation. Thus, the reviewers statement that the signals evaluated in this study cannot be attributed to the aerosol perturbation without a much larger perturbation, a large ensemble, or an O(1000 year) equilibrium simulation are not supported by the existing literature.

In addition, all of the below studies use comparable statistical tests to the one used here to attribute the signal seen in their simulations to the imposed aerosol perturbation. Indeed, several do not conduct the correction for autocorrelation conducted in this study. See for example:
Westervelt et al., 2020: statistical significance at the 95 % level according to a Student t test with the false discovery rate method from Wilks (2016) applied and an effective sample size adjusted for autocorrelation used [methodology comparable to that used in this study].
Westervelt et al. 2018: Hatching represents statistical significance at the 95 % level according to a Student's t test.
Kasoar et al., 2016: Stippling indicates that the change in that grid box exceeded 2 standard deviations [i.e. approximately the 95% confidence interval].
Kasoar et al., 2018: stippling indicates that the change at that grid-point exceeded 2 standard deviations [i.e. approximately the 95% confidence interval].
Liu et al., 2018: Stippled regions indicate where the multi-model mean change departs from zero by more than one standard deviation [i.e. approximately the 67% confidence interval – a weaker statistical threshold than the one applied here].
Samset et al., 2016: Hatched regions indicate where the multi-model mean is more than 1 standard deviation away from zero. [i.e. approximately the 67% confidence interval – a weaker statistical threshold than the one applied here]
Zhang et al., 2021: Hatching indicates where the changes are "significant" (90 % confidence).

References cited:
    Kasoar, M., Shawki, D. & Voulgarakis, A. Similar spatial patterns of global climate response to aerosols from different regions. *npj Climate and Atmospheric Science* **1**, 12 (2018).
    Kasoar, M. *et al.* Regional and global temperature response to anthropogenic $SO_2$ emissions from China in three climate models. *Atmospheric Chemistry and Physics* **16**, 9785–9804 (2016).

Westervelt, D. M. *et al.* Connecting regional aerosol emissions reductions to local and remote precipitation responses. *Atmospheric Chemistry and Physics* **18**, 12461–12475 (2018).

Westervelt, D. M. *et al.* Local and remote mean and extreme temperature response to regional aerosol emissions reductions. *Atmospheric Chemistry and Physics* **20**, 3009–3027 (2020).

Samset, B. H., G. Myhre, P. M. Forster, Ø. Hodnebrog, T. Andrews, G. Faluvegi, D. Fläschner, et al. 2016. "Fast and Slow Precipitation Responses to Individual Climate Forcers: A PDRMIP Multimodel Study." *Geophysical Research Letters* 43 (6): 2016GL068064. https://doi.org/10.1002/2016GL068064.

Santer, B. D., T. M. L. Wigley, J. S. Boyle, D. J. Gaffen, J. J. Hnilo, D. Nychka, D. E. Parker, and K. E. Taylor. 2000. "Statistical Significance of Trends and Trend Differences in Layer-Average Atmospheric Temperature Time Series." *Journal of Geophysical Research: Atmospheres* 105 (D6): 7337–56. https://doi.org/10.1029/1999JD901105.

Zhang, S., Stier, P. & Watson-Parris, D. On the contribution of fast and slow responses to precipitation changes caused by aerosol perturbations. *Atmos. Chem. Phys.* **21**, 10179–10197 (2021).

**Reviewer 2**

This is my second time viewing this paper and all my precius comments are satisfactory addressed.

I thank the reviewer for their favorable evaluation of the manuscript and for their earlier feedback, which contributed to improvements in the manuscript.